# Schauder Bases for $C[0,1]$ Using ReLU, Softplus and Two Sigmoidal Functions

**Anand Ganesh**                                                    *anandg@nias.res.in*
*National Institute of Advanced Studies,*
*Manipal Academy of Higher Education*

**Babhrubahan Bose**                                               *babhrubahanb@iisc.ac.in*
*Dept. of Mathematics*
*Indian Institute of Science, Bengaluru*

**Anand Rajagopalan**                                              *anandbr@gmail.com*
*Cruise*

**Reviewed on OpenReview:** *https://openreview.net/forum?id=YT79Qu1bOi*

## Abstract

We construct four Schauder bases for the space $C[0,1]$, one using ReLU functions, another using Softplus functions, and two more using sigmoidal versions of the ReLU and Softplus functions. This establishes the existence of a basis using these functions for the first time, and improves on the universal approximation property associated with them. We also show an $O(\frac{1}{n})$ approximation bound based on our ReLU basis, and a negative result on constructing multivariate functions using finite combinations of ReLU functions.

## 1 Introduction

Functions in spaces such as $C[0,1]$ with the supremum norm, $L^2[0,1]$, and $L^2(\mathbb{R})$ can be approximated, or in Kolmogorov's sense represented, using finite linear combinations of the form:

$$\sum_{i=1}^{N} \alpha_i \sigma(w_i x + b_i) \tag{1}$$

Expressions like equation 1 resemble single hidden-layer neural networks. The component functions $\sigma(w_i x + b_i)$, also known as plane waves, ridge functions, or sigmoids, have been widely applied in fields such as finance, data analysis, statistics, and medical imaging (Ismailov, 2021).

Although originally linked to Kolmogorov's representation theorem (Kolmogorov, 1957; Sprecher, 1965), this formulation has found broader utility in function approximation, as neatly delineated by Sprecher (Demb and Sprecher, 2021). This shift is seen in Cybenko's universal approximation theorem (Cybenko, 1989), where $\sigma$ is assumed to be a fixed sigmoidal function. The Kolmogorov-Sprecher representational approach uses function composition, like a two layer network, and the function $\sigma$ depends on the specific target function $f$. In contrast, Cybenko's approximation perspective is like a single layer network with no function composition, where $\sigma$ is fixed and independent of $f$.

Our work extends beyond these two perspectives by constructing infinite series representations using fixed activation functions. Specifically, we present four Schauder bases for $C[0,1]$: one based on the widely used ReLU function (Theorem 1), another based on its smooth variant, the Softplus function (Theorem 3), and two more based on sigmoidal versions of these functions (Theorem 2, Theorem 6).

There is a lot of interest in the neural network community on understanding the expressive power and approximation capabilities of ReLU based networks. Our basis results can be applied towards these questions. For instance, Theorem 7 shows an order $O(\frac{1}{n})$ approximation bound based on the first $n$ basis functions. This $O(\frac{1}{n})$ bound based on the supnorm on $C[0,1]$ is stronger than the $O(\frac{1}{n})$ $L^2$ bound in (Barron, 1993) since the $L^2$ norm on $C[0,1]$ is dominated by the supnorm. Further note that the $L^2$ error bound represents a sort of average or expected error while our result represents worst case error offering better protection against outliers. Our result is also an improvement, in terms of number of layers, on the construction of (Daubechies et al., 2022) for $L^2[0,1]$, as they require a multilayer network compared to our single layer basis or network, to obtain a $O(\frac{1}{n})$ $L^2$ approximation bound for univariate functions. In a limited way for univariate functions, our approximation result is also an improvement on Savarese et al. (2019) where they look at infinite width ReLU networks for functions $f : \mathbb{R} \to \mathbb{R}$. Whereas their $O(\frac{1}{n})$ $L^2$ error bound depends on target-specific basis functions, we achieve the same error bound with a fixed basis.

Finally we show a negative result in Theorem 8 that finite linear combinations of ReLU functions cannot represent multivariate functions in general, thus suggesting the need for deep neural networks to support multivariate functions. We hope to generalize this result to countable linear combinations, but this step remains open at this time.

In the context of neural networks, having a basis offers certain potential benefits compared to density results like those in Cybenko (1989). For example, a basis allows for structured initialization in neural network training. Consider an infinite expansion of the form:

$$f(x) = \sum_{i=1}^{\infty} \alpha_i \sigma_i(x), \quad \text{where } \sigma_i(x) = \sigma(w_i x + b_i)$$

This expansion is to be interpreted in the sense of a Schauder basis, as discussed in the next section. A finite truncation yields:

$$f(x) \approx f_M(x) = \sum_{i=1}^{M} \alpha_i \sigma_i(x)$$

Here, $f_M(x)$ can be seen as a neural network of width $M$. To improve the approximation, one could consider a wider network $f_N(x)$ with $N > M$, retaining both the functions $\sigma_i$ and coefficients $\alpha_i$ for $i \leq M$, and only learning the new coefficients for $i > M$. This reuse of parameters from a narrower model is theoretically justified only when an explicit basis is available. In contrast, density results offer no such structure and require retraining from scratch. Another possible advantage of using a basis is that it ensures unique expansions, and thus a unique global minimum during training.

In Banach space literature, functions similar to ReLU have appeared in basis constructions, such as the Schauder hat and restricted hat functions (Semadeni, 1982, p.28). However, the restricted hat, which is essentially a ReLU, is only used to describe the boundary behavior of a Schauder hat function, and not to construct a full basis. The ReLU and Softplus functions have not been previously employed for full basis constructions, possibly because their development was motivated by later applications in neural networks.

Within ridge function literature there is work on a universal sigmoidal function independent of the target function $f$. The existence of such a universal function for target functions in $C(\mathbb{R})$, with a prescribed approximation error, and a prescribed number of neurons is guaranteed by a theorem of Maiorov and Pinkus (Ismailov, 2021, p.158), and an algorithmic construction is provided by Ismailov (2021, p.164). Our basis construction (Theorem 1) seemingly provides such a universal function for $C[0,1]$, but this is misleading as the number of neurons will depend on the level of approximation desired. But again, the ability to approximate arbitrarily well with a fixed number of neurons depends on the use of wild and pathological functions, as Sprecher notes in Demb and Sprecher (2021), or the use of intricate algorithms as in Ismailov's smooth, almost monotone construction. In contrast, we use simple and standard functions like the ReLU in our basis constructions, and do not require the use of function composition.

From a theoretical standpoint, Cybenko (1989) notes that completeness results typically fall into two broad categories: those related to Weierstrass's theorem on polynomial density and those based on Wiener's

translation-invariant systems. In a way, our results incorporate aspects of both, much like Schauder's original basis from 1927.

Our construction, like Weierstrass's result, is situated within $C[0,1]$ and relies on a discrete bump function - the Schauder hat. At the same time, it employs scaled and shifted versions of a single activation function, akin to the translation-invariant methods of Wiener. This use of scaling and shifting is common in $L^2[0,1]$ literature, and in wavelet theory by extension.

## 2  Preliminaries

A countable sequence $\{x_n\}$ in a Banach space $X$ is a basis for $X$ if for all $x$ in $X$ there exist unique scalars $a_n(x)$ such that

$$x = \sum_{n=1}^{\infty} a_n(x)x_n \tag{2}$$

where the above series converges in the norm of $X$ (Heil, 2010). A normalized basis is a basis $\{x_n\}$ with $||x_n|| = 1$ for all $n$. A *Schauder* basis for $X$ is a basis for $X$ where the $a_n$'s are continuous linear functionals. There are other definitions of a Schauder basis that involve uncountable index sets, but we restrict ourselves to the above countable version. As it turns out, any basis for a Banach space is a Schauder basis. When $X = C[0,1]$ we use the standard supnorm topology. Since $C[0,1]$ is not a Hilbert space there is no notion of an inner product, and the linear functionals $a_n(x)$ are the closest approximation to the usual coordinates $\langle x_n, x \rangle$ based on an inner product $\langle ., . \rangle$.

The ReLU function $r(x)$ is defined as follows:

$$r(x) = \begin{cases} 0 & \text{if } x < 0, \\ x & \text{if } x \geq 0. \end{cases} \tag{3}$$

The parameterized Softplus function defined as follows (Dugas et al., 2000):

$$p_a(x) = \frac{\ln(1 + e^{ax})}{a}. \tag{4}$$

A function $\sigma : \mathbb{R} \to \mathbb{R}$ is called sigmoidal if

$$\lim_{x \to \infty} \sigma(x) = 1 \tag{5}$$

$$\lim_{x \to -\infty} \sigma(x) = 0 \tag{6}$$

Schauder's original basis functions for $C[0,1]$ ((Heil, 2010, p.142)) are defined by the ordered set $\mathcal{S} = \{s_{n,k} | n \in \mathbb{N} \cup \{0\}, k \in \{0, 1, \ldots, 2^n - 1\}\}$ under dictionary ordering, where

$$s_{n,k}(x) = \begin{cases} \frac{1}{2} & \text{if } x = \frac{k+\frac{1}{2}}{2^n}, \\ \text{linear} & \text{on } [\frac{k}{2^n}, \frac{k+\frac{1}{2}}{2^n}] \text{ and } [\frac{k+\frac{1}{2}}{2^n}, \frac{k+1}{2^n}], \\ 0 & \text{otherwise.} \end{cases} \tag{7}$$

Please note that $s_{n,k}(x) = \frac{1}{2}$ in the first case rather than $s_{n,k}(x) = 1$ as is standard. This change makes no difference to the basis property, but helps with some downstream calculations.

Given $f \in C[0,1]$ let us denote its basis expansion by:

$$f = \alpha_0 \chi_{[0,1]} + \alpha_1 s_1 + \sum_{k < 2^n} \alpha_{n,k} s_{n,k}. \tag{8}$$

Besides $\alpha_{n,k} s_{n,k}$, the Schauder basis expansion contains two additional terms $\alpha_0 \chi_{[0,1]}$ and $\alpha_1 s_1$. These involve two basis functions, namely the characteristic function $\chi_{[0,1]}$ and the linear function $s_1(x) = x$. We assume standard ordering for the Schauder basis, which is essentially a dictionary ordering of $(n, k)$. The ordering is important for convergence since the basis is conditional. We will now look at a basis construction for $C[0,1]$ based on the above ReLU function $r(x)$.

## 3 ReLU Basis

We now construct a Schauder basis for $C[0,1]$ using the ReLU function $r(x)$ and two auxiliary functions. In particular, we prove the following theorem:

**Theorem 1.** *The basis functions* $\chi_{[0,1]}(x)$, $s_1(x) = x$, $r(2^n x - k)$ *and* $r(2^n x - (k + \frac{1}{2}))$ *form a Schauder basis for* $C[0,1]$. *In particular, borrowing* $\alpha_{n,k}$ *from the Schauder basis expansion equation 7, and setting* $\alpha_{n,-1} = 0$, *we have*

$$f = \alpha_0 \chi_{[0,1]} + \alpha_1 s_1 + \sum_{n=0}^{\infty} \sum_{k=0}^{2^n-1} \{(\alpha_{n,k} + \alpha_{n,k-1})r(2^n x - k) - 2\alpha_{n,k}r(2^n x - (k + \frac{1}{2}))\}. \tag{9}$$

*where the coefficient functionals* $\alpha_0$, $\alpha_1$, $(\alpha_{n,k} + \alpha_{n,k-1})$ *and* $-2\alpha_{n,k}$ *are all bounded.*

Please note that we have simplified notation by implicitly restricting our basis functions to $[0,1]$, and we will continue to do so. i.e. we will write $f(x)$ for $f(x)|_{[0,1]}$.

We begin our proof with the following lemma:

**Lemma 1.** *Define* $t_{n,k}$ *as follows:*

$$t_{n,k}(x) = r(2^n x - k) - 2r(2^n x - (k + \frac{1}{2})) + r(2^n x - (k+1)). \tag{10}$$

. *We claim that* $t_{n,k}(x) = s_{n,k}(x)$ *for all* $x \in \mathbb{R}$.

The proof of this lemma is entirely routine. The heart of Theorem 1 is in the specific construction of $t_{n,k}$ and the proof of boundedness which follows the proof of Lemma 1 below. Not all linear combinations similar to $t_{n,k}$ lead to valid Schauder basis. For instance, consider $g_{n,k}(x) = r(2^n x - k) - r(2^n x - (k + \frac{1}{2})) - r(-2^n x + (k + \frac{1}{2})) + r(-2^n x + (k+1)) - \frac{1}{2}$. We can prove that $g_{n,k}(x) = s_{n,k}(x)$, but unlike $t_{n,k}$, $g_{n,k}$ does not lead to a ReLU basis. The boundedness proof of Theorem 1 does not work for $g_{n,k}$.

*Proof.* Let $x = \frac{k+\delta}{2^n}$. Then,

$$\begin{aligned} t_{n,k} &= r(2^n \frac{k+\delta}{2^n} - k) - 2r(2^n \frac{k+\delta}{2^n} - (k + \frac{1}{2})) + r(2^n \frac{k+\delta}{2^n} - (k+1)) \\ &= r(\delta) - 2r(\delta - \frac{1}{2}) + r(\delta - 1). \end{aligned}$$

We will now establish equality of $t_{n,k}$ and $s_{n,k}$ in all the four cases listed in equation 7.

For $x = \frac{k+\frac{1}{2}}{2^n}$, we have $\delta = \frac{1}{2}$

$$\begin{aligned} t_{n,k} &= r(\delta) - 2r(\delta - \frac{1}{2}) + r(\delta - 1) \\ &= r(\frac{1}{2}) - 2r(0) + r(\frac{-1}{2}) \\ &= r(\frac{1}{2}) \\ &= \frac{1}{2}. \end{aligned}$$

For $x \in [\frac{k}{2^n}, \frac{k+\frac{1}{2}}{2^n}]$, $\delta \in [0, \frac{1}{2}]$. Thus,

$$t_{n,k} = r(\delta) = \delta = 2^n x - k,$$

which is linear in $x$.

Similarly, for $x \in [\frac{k+\frac{1}{2}}{2^n}, \frac{k+1}{2^n}]$, we have $\delta \in [\frac{1}{2}, 1]$. So,

$$t_{n,k} = r(\delta) - 2r(\delta - \frac{1}{2}) + r(\delta - 1)$$
$$= \delta - 2(\delta - \frac{1}{2}) + 0$$
$$= 1 - \delta = \frac{1}{2} - (2^n x - k)$$
$$= k + 1 - 2^n x,$$

which is also linear in $x$.

Next for $x < \frac{k}{2^n}$, $\delta \leq 0$. Thus,

$$t_{n,k} = r(\delta) - 2r(\delta - \frac{1}{2}) + r(\delta - 1)$$
$$= 0$$

Finally, for $x > \frac{k+1}{2^n}$, $\delta > 1$ and

$$t_{n,k} = r(\delta) - 2r(\delta - \frac{1}{2}) + r(\delta - 1)$$
$$= \delta - 2(\delta - \frac{1}{2}) + (\delta - 1)$$
$$= 0.$$

We have $t_{n,k}(x) = s_{n,k}(x)$ in all the cases considered, and thus $t_{n,k}(x) = s_{n,k}(x)$ for all $x \in \mathbb{R}$.

$\square$

We will now prove Theorem 1 on the boundedness of the coefficient functionals.

*Proof.*

$$f = \alpha_0 \chi_{[0,1]} + \alpha_1 s_1 + \sum_{n=0}^{\infty} \sum_{k=0}^{2^n-1} \alpha_{n,k} s_{n,k}$$

$$= \alpha_0 \chi_{[0,1]} + \alpha_1 s_1 + \sum_{n=0}^{\infty} \sum_{k=0}^{2^n-1} \alpha_{n,k} t_{n,k}$$

$$= \alpha_0 \chi_{[0,1]} + \alpha_1 s_1 + \sum_{n=0}^{\infty} \sum_{k=0}^{2^n-1} \alpha_{n,k} \{r(2^n x - k) - 2r(2^n x - (k + \frac{1}{2})) + r(2^n x - (k+1))\}$$

$$= \alpha_0 \chi_{[0,1]} + \alpha_1 s_1 + \sum_{n=0}^{\infty} \sum_{k=0}^{2^n-1} \{\alpha_{n,k} r(2^n x - k) - 2\alpha_{n,k} r(2^n x - (k + \frac{1}{2})) + \alpha_{n,k} r(2^n x - (k+1))\}$$

$$= \alpha_0 \chi_{[0,1]} + \alpha_1 s_1 + \sum_{n=0}^{\infty} \{\alpha_{n,0} r(2^n x) + \sum_{k=1}^{2^n-1} \{(\alpha_{n,k} + \alpha_{n,k-1}) r(2^n x - k) - 2\alpha_{n,k} r(2^n x - (k + \frac{1}{2}))\}\} \quad (*)$$

$$= \alpha_0 \chi_{[0,1]} + \alpha_1 s_1 + \sum_{n=0}^{\infty} \sum_{k=0}^{2^n-1} \{(\alpha_{n,k} + \alpha_{n,k-1}) r(2^n x - k) - 2\alpha_{n,k} r(2^n x - (k + \frac{1}{2}))\}$$

where, for convenience, we have set $\alpha_{n,-1} = 0$ in the last step. Notice that equation $*$ does not involve any rearrangement of terms. The grouping of $(\alpha_{n,k} + \alpha_{n,k-1})$ involves only associativity, and no commutativity. In particular, the conditional convergence of the earlier series and that of equation $*$ are equivalent.

Finally, given that the coefficient functionals $\alpha_{n,k}$ are bounded, the coefficients of the ReLU expansion, namely, $\alpha_0$, $\alpha_1$, $\alpha_{n,k} + \alpha_{n,k-1}$ and $-2\alpha_{n,k}$ are all bounded as well. This establishes that the sequence of functions $\chi_{[0,1]}(x)$, $s_1(x) = x$, $r(2^n x - k)$ and $r(2^n x - (k + \frac{1}{2}))$ form a Schauder basis. □

*Remarks:* Let $t(x)$ be defined as follows to be the discrete second derivative of $r(x)$:

$$t(x) = r(x) - 2r(x - \frac{1}{2}) + r(x - 1). \tag{11}$$

$t(x)$ is a triangular bump or hat function which is the essential building block of the original basis of Schauder. Letting $d(x)$ denote the expression $r(x) - r(x - \frac{1}{2})$, we can see that $t(x)$ matches the expression $d(x) - d(x - \frac{1}{2})$. Here $d(x)$ can be interpreted as the first discrete derivative of $r(x)$, and $t(x)$ as the corresponding first discrete derivative of $d(x)$, or in effect the second discrete derivative of $r(x)$.

The subscripted functions $r_{n,k}(x)$, $d_{n,k}(x)$ and $t_{n,k}(x)$ are dyadically scaled and shifted versions of $r(x)$, $d(x)$ and $t(x)$, for instance $t_{n,k}(x) = t(2^n x - k)$. Correspondingly, $d_{n,k}(x)$ is the first discrete derivative of $r_{n,k}(x)$ and $t_{n,k}(x)$ is the first discrete derivative of $d_{n,k}(x)$, or in effect the second discrete derivate of $r_{n,k}(x)$. It is well known that the Haar basis elements represent the first derivative of the Schauder basis elements. The current construction shows that the Schauder basis elements represent the first discrete derivative of $d_{n,k}$ and the second discrete derivative of the ReLU basis elements $r_{n,k}$.

$d_{n,k}(x)$ is a continuous sigmoidal function, and thus dense in $C[0,1]$ as per Cybenko (1989). Going beyond Cybenko's result, $d_{n,k}$ can be used to assemble a basis as shown in the following theorem:

**Theorem 2.** *The functions $d_{n,k}(x)$ with $n \geq 0$ and $k \leq 2^n - 1$ along with the auxiliary functions $\chi_{[0,1]}(x)$ and $s_1(x) = 2$ forms a Schauder basis for $C[0,1]$ with expansions of the following form for all $f$ in $C[0,1]$:*

$$f = \alpha_0 \chi_{[0,1]} + \alpha_1 s_1 + \sum_{n=0}^{\infty} \sum_{k=0}^{2^n-1} \alpha_{n,k} d_{n,k} - \alpha_{n,k} d_{n,k-\frac{1}{2}} \tag{12}$$

*where the coefficient functionals are all bounded.*

*Proof.* We start with the basis expansion of $f$ using the basis elements $t_{n,k}$ where $\alpha_j$ and $\alpha_{n,k}$ are borrowed from the Schauder basis expansion equation 7.

$$f = \alpha_0 \chi_{[0,1]} + \alpha_1 s_1 + \sum_{n=0}^{\infty} \sum_{k=0}^{2^n-1} \alpha_{n,k} t_{n,k}$$

$$= \alpha_0 \chi_{[0,1]} + \alpha_1 s_1 + \sum_{n=0}^{\infty} \sum_{k=0}^{2^n-1} \alpha_{n,k} (d_{n,k} - d_{n,k-\frac{1}{2}})$$

$$= \alpha_0 \chi_{[0,1]} + \alpha_1 s_1 + \sum_{n=0}^{\infty} \sum_{k=0}^{2^n-1} \alpha_{n,k} d_{n,k} - \alpha_{n,k} d_{n,k-\frac{1}{2}}$$

This concludes the proof since the coefficient functionals $\alpha_0$, $\alpha_1$ and $\alpha_{n,k}$ are all known to be bounded. □

We note a general principle at play here with regards to first and second discrete derivatives. In particular, if $f_j \in X$ form a Schauder basis for Banach space $X$, and if $f_j$ are the first discrete derivatives of $g_j$ and the second discrete derivatives of $h_j$, then $g_j$ and $h_j$ form Schauder bases for $X$ as well. One may argue that the basis property of $h_j$ follows from the basis property of $g_j$ by induction, but an examination of the above proofs shows that some care is required to avoid double counting as seen in the functional $(\alpha_{n,k} + \alpha_{n,k-1})$.

As noted earlier, $d_{n,k}$ forms a basis, but $d(x)$ is not a universal sigmoidal function for $C[0,1]$. Unlike the construction of Ismailov (2021, p.164) for $C(\mathbb{R})$, the number of terms in a series truncation increases with the desired level of approximation.

## 4 Softplus Basis

Schauder bases possess some stability properties wherein the basis property holds even if each element is perturbed slightly. We use this stability property to perturb the ReLU basis and obtain a basis using Softplus functions. In particular, we will prove the following theorem:

**Theorem 3.** *The basis functions* $\chi_{[0,1]}(x)$, $s_1(x) = x$, $p_{a(n,k)}(2^n x - k)$ *and* $p_{a(n,k)}(2^n x - (k + \frac{1}{2}))$ *form a Schauder basis for* $C[0,1]$. *In particular, given* $f$ *in* $C[0,1]$, *we have the following basis expansion with* $a(n,k) = 4 \ln 2 \cdot 2K \cdot 2^{2n+2}$

$$f = \gamma_0 \chi_{[0,1]} + \gamma_1 s_1 + \sum_{n=0}^{\infty} \sum_{k=0}^{2^n - 1} \{\gamma_{n,k} p_{a(n,k)}(2^n x - k) + \psi_{n,k} p_{a(n,k)}(2^n x - (k + \frac{1}{2}))\}. \tag{13}$$

*where the coefficient functionals* $\gamma_0$, $\gamma_1$, $\gamma_{n,k}$ *and* $\psi_{n,k}$ *are all bounded.*

We start by recalling some stability properties of Schauder bases. First, we have the notion of a basis constant whose existence is asserted in the following classical theorem.

**Theorem 4.** *(Lindenstrauss and Tzafriri, 1977) Let* $\{x_n\}$ *be a Schauder basis of a Banach Space* $X$. *Then the projections* $P_n : X \to X$ *defined by* $P_n(\sum_{i=1}^{\infty} a_i x_i) = \sum_{i=1}^{n} a_i x_i$ *are bounded linear operators and* $\sup_n \|P_n\| < \infty$. *The supremum* $K = \sup_n \|P_n\|$ *is called the basis constant of* $\{x_n\}$.

We then have the following stability property:

**Theorem 5.** *(Lindenstrauss and Tzafriri, 1977, Prop. 1.a.9) Let* $\{x_n\}$ *be a normalized Schauder basis of a Banach Space* $X$ *with basis constant* $K$. *If* $\{y_n\}$ *is a sequence of vectors in* $X$ *such that* $\sum_{n=1}^{\infty} \|x_n - y_n\| < \frac{1}{2K}$, *then* $\{y_n\}$ *is also a Schauder basis of* $X$. *This property holds for any fixed normalization constant* $c > 0$ *such that* $\|x_n\| = c \ \forall \ n$.

To prove that we can construct a basis with Softplus functions, we will start with an intermediate basis whose basis elements $q_{n,k}$ are defined as follows using the parameterized Softplus function:

$$q_{n,k}(x) = p_a(2^n x - k) - 2p_a(2^n x - (k + \frac{1}{2})) + p_a(2^n x - (k + 1)) \tag{14}$$

$$a = a(n,k) = 4 \ln 2 \cdot 2K \cdot 2^{2n+2}. \tag{15}$$

We then have the following lemma:

**Lemma 2.** *The functions* $\chi_{[0,1]}(x)$, $s_1(x) = x$ *and* $q_{n,k}(x)$ $(n \geq 0, 0 \leq k \leq 2^n - 1)$ *form a Schauder basis for* $C[0,1]$.

*Proof.* Let us apply the stability property in Theorem 5 with $X = C[0,1]$ to perturb the ReLU basis $t_{n,k}$. Let $K$ denote the basis constant for the ReLU basis and note that the ReLU basis is a normalized basis with $\|t_{n,k}\| = 1$ as required by Theorem 5.

We will perturb the ReLU basis elements $t_{n,k}$ using a parameterized Softplus function $p_a(x)$ instead of the ReLU function $r(x)$ to obtain our new basis elements $q_{n,k}$. As we show below, increasing $a$ as a suitable multiple of $2^{2n+2}$, will ensure that the individual perturbations get smaller, and that the total perturbation across all basis elements remains small.

A simple analysis of the parameterized Softplus function $p_a(x)$ equation 4 shows that $|p_a(x) - r(x)|$ attains its maximum at $x = 0$, and this maximum value is $p_a(0) = \frac{\ln 2}{a}$. This maximum value drops as $\frac{1}{a}$ when we increase the sharpness parameter $a$.

We further observe that for $a$ fixed, $\sup |p_a(2^n x - k) - r(2^n x - k)| = \sup |p_a(x) - r(x)| = \frac{\ln 2}{a}$. That is, the supremum of the difference between $p_a$ and $r$ does not change when the function parameters are scaled and shifted. Since $\sup_x |p_a(x) - r(x)|$ equals $\|p_a - r\|$, we can see that the norm of the perturbation $\|p_a - r\|$

remains unchanged for a given $a$ even when the function parameters for $p_a(x)$ and $r(x)$ are scaled and shifted. This gives us the following bound on the perturbation error between individual basis elements $t_{n,k}$ and $q_{n,k}$:

$$
\begin{aligned}
||t_{n,k} - q_{n,k}|| &= \sup |(r(2^n x - k) - 2r(2^n x - (k + \tfrac{1}{2})) + r(2^n x - (k + 1))) \\
&\quad - (p_a(2^n x - k) - 2p_a(2^n x - (k + \tfrac{1}{2})) + p_a(2^n x - (k + 1)))| \\
&\leq \sup |r(2^n x - k) - p_a(2^n x - k)| + 2|r(2^n x - (k + \tfrac{1}{2})) - p_a(2^n x - (k + \tfrac{1}{2}))| \\
&\quad + |r(2^n x - (k + 1)) - p_a(2^n x - (k + 1))| \\
&= \sup |r(x) - p_a(x)| + 2|r(x) - p_a(x)| + |r(x) - p_a(x)| \\
&= 4 \sup |r(x) - p_a(x)| \\
&= \frac{4 \ln 2}{a}.
\end{aligned}
$$

The total perturbation error $\Delta$ across all basis elements can now be bounded as:

$$
\begin{aligned}
\Delta &= \sum_{n=0}^{\infty} \sum_{k=0}^{2^n - 1} ||t_{n,k} - q_{n,k}|| \\
&\leq \sum_{n=0}^{\infty} \sum_{k=0}^{2^n - 1} \frac{4 \ln 2}{a} \\
&\leq \sum_{n=0}^{\infty} \sum_{k=0}^{2^n - 1} \frac{1}{2K} \frac{1}{2^{n+2}} \frac{1}{2^n} \\
&= \frac{1}{2K} \sum_{n=0}^{\infty} \frac{1}{2^{n+2}} \\
&< \frac{1}{2K}
\end{aligned}
$$

Since the total perturbation $\Delta < \frac{1}{2K}$, we conclude that $q_{n,k}$ is a Schauder basis for $C[0,1]$. This proves Lemma 2. $\qquad \square$

Notice that $q_{n,k}$ form a smooth basis, but the elements of this basis are not scaled and shifted versions of a single mother function. The sharpness parameter destroys this scale-shift property of the basis. We also note that this construction cannot be based on sigmoidal functions like tanh since they lack a sharpness parameter to control the supnorm error, and to thus bound the total perturbation.

We will now prove Theorem 3 establishing that $p_a(x)$ forms a basis. We mimic the proof of Theorem 1 though we make use of the basis elements $q_{n,k}$ defined above instead of the original Schauder basis elements $s_{n,k}$.

*Proof.* We will start with a basis expansion using $q_{n,k}$ denoting the coefficient functionals as $\beta_j$ and $\beta_{n,k}$:

$$f = \beta_0 \chi_{[0,1]} + \beta_1 s_1 + \sum_{n=0}^{\infty} \sum_{k=0}^{2^n-1} \beta_{n,k} q_{n,k}$$

$$= \beta_0 \chi_{[0,1]} + \beta_1 s_1 + \sum_{n=0}^{\infty} \sum_{k=0}^{2^n-1} \beta_{n,k} \{ p_a(2^n x - k) - 2 p_a(2^n x - (k + \tfrac{1}{2})) + p_a(2^n x - (k+1)) \}$$

$$= \beta_0 \chi_{[0,1]} + \beta_1 s_1 + \sum_{n=0}^{\infty} \sum_{k=0}^{2^n-1} \{ \beta_{n,k} p_a(2^n x - k) - 2 \beta_{n,k} p_a(2^n x - (k + \tfrac{1}{2})) + \beta_{n,k} p_a(2^n x - (k+1)) \}$$

$$= \beta_0 \chi_{[0,1]} + \beta_1 s_1 + \sum_{n=0}^{\infty} \{ \beta_{n,0} p_a(2^n x) + \sum_{k=1}^{2^n-1} \{ (\beta_{n,k} + \beta_{n,k-1}) p_a(2^n x - k) - 2 \beta_{n,k} p_a(2^n x - (k + \tfrac{1}{2})) \} \} \quad (*)$$

$$= \beta_0 \chi_{[0,1]} + \beta_1 s_1 + \sum_{n=0}^{\infty} \sum_{k=0}^{2^n-1} \{ (\beta_{n,k} + \beta_{n,k-1}) p_a(2^n x - k) - 2 \beta_{n,k} p_a(2^n x - (k + \tfrac{1}{2})) \}$$

where, for convenience, we have set $\beta_{n,-1} = 0$ in the last step. Like in the earlier proof, we note that equation $*$ preserves conditional convergence. Finally, given that the coefficient functionals $\beta_j$ and $\beta_{n,k}$ are bounded, the coefficients of the Softplus expansion, namely, $\gamma_0 = \beta_0$, $\gamma_1 = \beta_1$, $\gamma_{n,k} = (\beta_{n,k} + \beta_{n,k-1})$ and $\psi_{n,k} = -2\beta_{n,k}$ are all bounded as well. This establishes that the sequence of functions $\chi_{[0,1]}(x)$, $s_1(x) = x$, $p_a(2^n x - k)$ and $p_a(2^n x - (k + \frac{1}{2}))$ form a Schauder basis, and concludes the proof of Theorem 3. $\square$

As a simple corollary, we construct a sigmoidal basis based on Softplus functions. The construction of $u_{n,k}$ mimics the construction of the first discrete derivative $d_{n,k}$ in Theorem 2.

**Theorem 6.** *The functions $u_{n,k}(x)$ defined to be $p_{a(n,k)}(2^n x - k) - p_{a(n,k)}(2^n x - (k + \frac{1}{2}))$, with $a(n,k) = 4 \ln 2 \cdot 2K \cdot 2^{2n+2}$, for $n \geq 0, 0 \leq k \leq 2^n - 1$, along with the auxiliary functions $\chi_{[0,1]}(x)$ and $s_1(x) = x$ form a sigmoidal Schauder basis for $C[0,1]$ .*

Observe that $u_{n,k}$ are indeed sigmoidal functions since $\lim_{x \to -\infty} u_{n,k}(x)$ equals 0, and $\lim_{x \to \infty} u_{n,k}(x)$ equals 1. It is also easy to see that $u_{n,k}$ are smooth, monotonically increasing functions. We skip the proof of the basis expansion as it follows the same lines as Theorem 2 except we use $q_{n,k}$ for the initial basis expansion instead of $t_{n,k}$.

## 5   Applications

In machine learning theory, it is well known that in the univariate case, one can achive $O(\frac{1}{N})$ approximation error with $N$ functions. We prove that our basis expansions achieve the same order of approximation with $N$ functions based on a simple interpolation scheme. Setting $r_{n,k}(x) = r(2^n x - k)$, have the following:

**Theorem 7.** *Given a Lipschitz function $f \in C[0,1]$ with Lipschitz constant $c > 0$, and a basis expansion $f = \alpha_0 \chi_0 + \alpha_1 \chi_1 + \sum_{n=0}^{\infty} \sum_{k=0}^{n} (\alpha_{n,k} r_{n,k} + \beta_{n,k} r_{n,k+\frac{1}{2}})$, we can obtain $O(\frac{1}{N})$ approximations using the first $N$ basis functions. In particular, given the first $N$ basis functions referred to in general as $b_i$, there exist $\beta_i$ such that $\|f - \sum_{i=0}^{N} \beta_i b_i\| \leq \frac{K}{N}$ for $N$ sufficiently large, and for some fixed constant $K$ independent of $N$.*

*Proof.* The argument is a standard one for Lipschitz functions based on interpolation of equispaced points. Let us first set $\alpha_0 = f(0)$ and $\alpha_1 = f(1) - f(0)$ where $\chi_0(x) = 1$, the constant function and $\chi_1(x) = x$, the identify function. We see that the function $g = f - (\alpha_0 \chi_0 + \alpha_1 \chi_1)$ has $g(0) = 0$ and $g(1) = 0$. Based on this simple device, for the rest of this proof, we will assume without loss of generality that $f(0) = f(1) = 0$ and do our interpolation based on the other basis functions without using $\chi_0$ and $\chi_1$.

We will derive a $O(\frac{1}{N})$ approximation error based on the Schauder basis $s_{n,k}(x)$, and use a simple counting argument to prove that the same $O(\frac{1}{N})$ approximation error carries over to the ReLU basis as well.

Given $N > 0$, let $P = 2^p$ be the largest power of 2 less than or equal to $N$. Clearly $P \geq \frac{N}{2}$ Notice that the first $P$ terms of the basis expansion include all the basis functions of the form $s_{2^p,k}$ for $k \in [0, 2^p]$. The peaks of these basis functions are the dyadic points $x_1 = \frac{1}{2^{p+1}}, x_2 = \frac{3}{2^{p+1}}, \ldots, x_k = \frac{2k+1}{2^{p+1}}$ for $k \leq 2^p - 1$. Let us now build an interpolating function $f_P$ as $f_P(x) = \sum 2f(x_k)s_{2^p,k}(x)$. It is clear that $f_P(x_k) = f(x_k) \; \forall \; k$ since the support of each basis function $s_{2^p,k}$, essentially the hat, is limited to the interval $[x_k - \frac{1}{2^{p+1}}, x_k + \frac{1}{2^{p+1}}]$. Further, $f_P$ is piecewise linear in $(x_k - \frac{1}{2^{p+1}}, x_k + \frac{1}{2^{p+1}})$, and for $x \in (x_k - \frac{1}{2^{p+1}}, x_k + \frac{1}{2^{p+1}})$, and $x \neq x_k$, we have $f'_P(x) \leq c$, where $c$ is the Lipschitz constant of $f$. This immediately gives us the approximation bound we are looking for:

$$\begin{aligned}
|f_P(x) - f(x)| &= |f_P(x) - f(x_k) + f(x_k) - f(x)| \\
&= |f_P(x) - f_P(x_k) + f(x_k) - f(x)| \\
&\leq |f_P(x) - f_P(x_k)| + |f(x_k) - f(x)| \\
&\leq \frac{2c}{P} \leq \frac{2c}{\frac{N}{2}} = O(\frac{1}{N})
\end{aligned}$$

as required for the Schauder basis $s_{n,k}$. Now, we note that $P$ Schauder basis functions $s_{n,k}$ are subsumed by $2 * P$ ReLU basis functions $r(2^n x - k), r(2^n x - k + \frac{1}{2})$. In other words, the $N$-term approximation error for ReLU functions is bounded by $\frac{2c}{\frac{P}{2}} = O(\frac{1}{N})$ as required. $\qquad \square$

We end with a simple but intriguing proposition that suggests the power of depth in ReLU networks. The proposition is a negative result that our basis property does not generalize to dimensions $d \geq 2$, and that multilayer networks are required to represent multivariate functions in general. We do not have a proof of this proposition yet, but we offer a preliminary result involving finite sums instead of countable sums that conveys the general idea, and we believe this idea can be generalized. This negative proposition ties in naturally with the Kolmogorov-Sprecher representation theorems which require function composition, in effect 2-layer networks. It is interesting future work to construct such a 2-layer or multilayer network, perhaps based on the pyramidal functions of (Semadeni, 1982).

Generalizing the theorem below from finite linear combinations to countable linear combinations will prove the actual proposition we are looking for, that one cannot have a Schauder basis for $C[0, 1]^d$ based on ReLU functions.

**Theorem 8.** *There exist functions in $C[0, 1]^d$ that cannot be represented using a finite linear combination of ReLU functions of the form $r(w^T x + b)$.*

*Proof.* We will exhibit a pyramidal function $f : [0, 1]^2 \to \mathbb{R}$ that cannot be expressed as a finite combination of ReLU functions. From this, we can obtain similar negative results for $f_d : [0, 1]^d \to \mathbb{R}$ with $d > 2$, by setting $f_d(x_1, x_2, \ldots, x_d) := f(x_1, x_2)$. To verify this, let us assume, for the sake of contradiction, that $f_d$ can be expressed as a finite sum of ReLU functions as $f_d(x_1, x_2, \ldots, x_d) = \sum_{i=1}^{N} \alpha_i r \left( w_i^T (x_1, x_2, \ldots, x_d) + b_i \right)$. In that case, setting $x_3 = \cdots = x_d = 0$, we get $f(x_1, x_2) = \sum_{i=1}^{N} \alpha_i r \left( w_i'^T (x_1, x_2) + b_i \right)$ where $w_i'$ denotes the vector consisting of the first two coordinates of $w_i$. This violates the assumption that $f$ cannot be represented as a finite linear combination of ReLU functions.

Now consider a pyramidal function $f : [0, 1]^2 \to \mathbb{R}$ (as visualized in Figure 4, Appendix A) with peak value of 1 at $(\frac{1}{2}, \frac{1}{2})$ and zero on and outside the base. The base, denoted as $B$, is given by the set of points enclosed by the lines $x = \frac{1}{4}$, $x = \frac{3}{4}$, $y = \frac{1}{4}$, $y = \frac{3}{4}$, or equivalently as the convex hull of $\{(\frac{1}{4}, \frac{1}{4}), (\frac{3}{4}, \frac{1}{4}), (\frac{3}{4}, \frac{3}{4}), (\frac{1}{4}, \frac{3}{4})\}$. We claim that this pyramidal function $f$ cannot be constructed using a finite number of ReLU functions. Suppose to the contrary, that $f(x) = \sum_{i=1}^{N} \alpha_i r_i(x)$ where $x \in [0, 1]^2$ and $r_i(x) = r(w_i^T x + b_i)$. Without loss of generality, we assume that none of the $w_i$ or $\alpha_i$ are zero, and the tuples $(w_i, b_i)$ are distinct.

For a given $r_i$ define the associated line $l_i$ by $\{x \in [0, 1]^2 \mid w_i^T x + b_i = 0\}$. Notice that for each of the lines $l_i$, $r_i$ is zero (or *off*) on one side of the line, and affine (or *on*) on the other side. Note that the $l_i$'s along with the four boundaries of the unit square partition $[0, 1]^2$ into convex polygonal regions (which we take to be closed), and we denote the set of these regions as $\mathcal{R}$. Given two regions $R, Q \in \mathcal{R}$, we call them neighbors if

they share some part of an edge. A region $R \in \mathcal{R}$ is called a zero region if $f(x) = 0 \ \forall \ x \in R$, and a non-zero region otherwise.

Our main observation is that a zero region cannot have another zero region as a neighbor. To see this, let us first define $z_i : [0,1]^2 \to \{0,1\}$ as $z_i(x) = 0$ whenever $r_i(x) \leq 0$ and $z_i(x) = 1$ whenever $r_i(x) > 0$. $z_i(x)$ is a binary valued function which tracks whether $r_i$ is *on* or *off*, and thus $r_i(x) = z_i(x)(w_i^T x + b_i)$. For a given region $R \in \mathcal{R}$ and a fixed $i$, $z_i(x)$ is the same for all $x \in R$. It is clear that crossing the line $l_i$ changes the value of $z_i(x)$ from zero to one or vice versa, and that $z_j$ remains unchanged for $j \neq i$.

Let us now consider the weight matrix $w(x) = \sum_{i=1}^{N} \alpha_i z_i(x) w_i^T x$ and bias matrix $b(x) = \sum_{i=1}^{N} \alpha_i z_i(x) b_i$. Within a given region $R \in \mathcal{R}$, $w(x)$ and $b(x)$ remain constant since $z_i(x)$ remains constant. Let us denote by $w_R$, the constant weight matrix for $R$, and by $b_R$, the constant bias matrix. Now we note that $w_R$ must be zero for a zero region $R$. Indeed, if $w_R$ were not zero, we can find $x \in R$ such that $w_R^T x + b_R \neq 0$. Now, given neighboring regions $Q, R \in \mathcal{R}$ with a common edge $l_i$, $w_Q = w_R \pm \alpha_i w_i(x)$ which implies that $w_R$ and $w_Q$ cannot both be zero since $w_i, \alpha_i \neq 0$. This verifies the main observation above that a zero region cannot have another zero region as a neighbor.

Let us now consider one of the corner points $P$ of the base of the pyramid $B$, and all the regions outside $B$ which contain $P$ as a vertex. There must be at least two such regions since the regions are convex, and they subtend a total angle of $\frac{3\pi}{2}$ at $P$. But all these regions must be zero regions since $f$ is identically zero outside $B$, which forces two adjacent zero regions, contradicting our main observation.

Thus, the pyramidal function $f$ cannot be represented as finite linear combination of the form $f(x) = \sum_{i=1}^{N} \alpha_i r_i(x)$.

$\square$

Note that the method of the proof can be used to generalize the above theorem to pyramidal functions whose base is a convex polygon strictly contained in the unit square.

## 6 Conclusion

We constructed four Schauder bases for $C[0,1]$, one using the ReLU function (Theorem 1), another using the Softplus function (Theorem 3), and two more using sigmoidal versions of the above (Theorem 2 and Theorem 6). The last basis consists of smooth, monotonically increasing sigmoidal functions. Finally we show an $O(\frac{1}{n})$ approximation bound using our ReLU basis (Theorem 7) and a negative result on constructing multivariate functions with finite linear combinations of ReLU functions (Theorem 8).

In terms of future work, we wonder if scaled and shifted bases for $C[0,1]$ are possible using smooth functions. In particular, we pose the following question: does there exist a smooth function $\sigma : [0,1] \to \mathbb{R}$ such that $\sigma_{n,k}(x) = \sigma(2^n x - k)$ forms a basis for $C[0,1]$ ?

**Acknowledgement.** Dedicated to *thatha-patti*. Our sincere thanks to Prof. Nithin Nagaraj for supporting this work, to Prof. K B Sinha for early comments on problem formulation, and to the anonymous reviewers for the time spent, and helpful pointers connecting our mathematical results to machine learning theory.

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

# A  Function Plots

This section contains function plots for easy visualization. We start with the ReLU function $r(x)$, then look at the first differences of these ReLU function $d_{n,k}(x)$, and then the second differences $t_{n,k}(x)$ which correspond to the Schauder hat functions $s_{n,k}(x)$ as well. The last row contains the main part of the counter example $g_{n,k}(x)$, then a perturbed Schauder hat function built from softplus functions, and finally the pyramidal function used in Theorem 8.

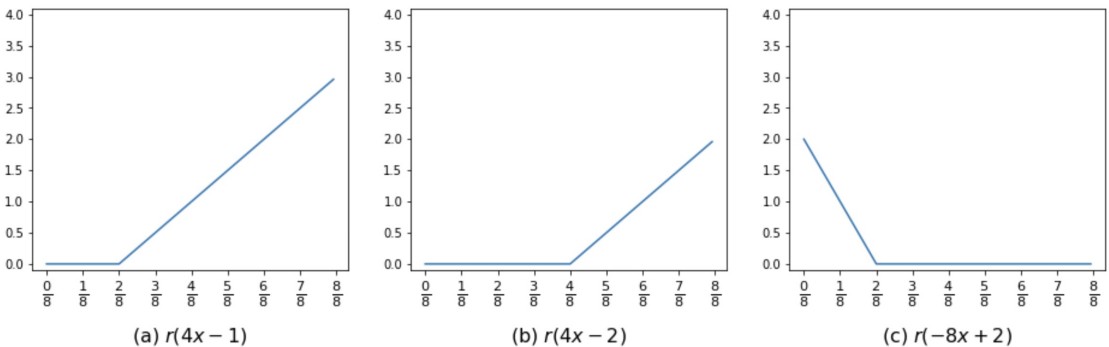

Figure 1: ReLU function plots

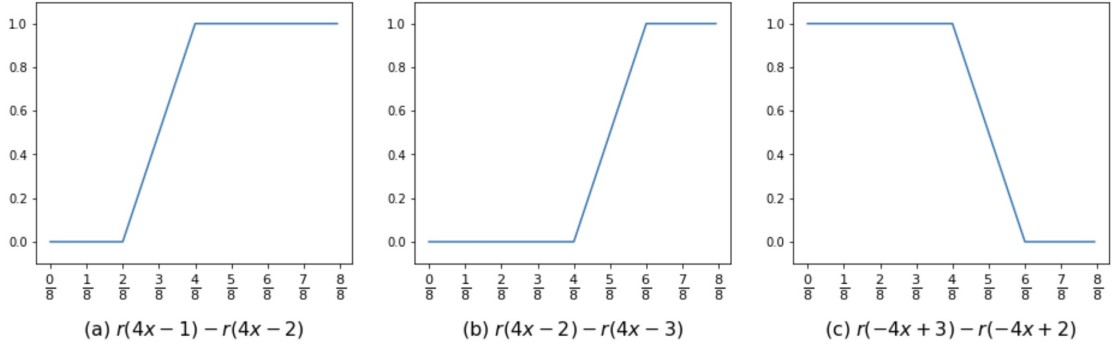

Figure 2: First Differences $d_{n,k}(x)$

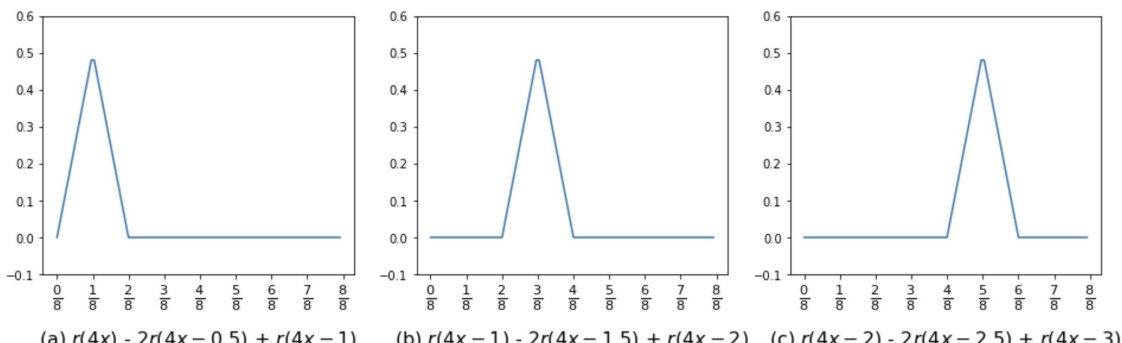

Figure 3: Second Differences $t_{n,k}(x) = s_{n,k}(x)$

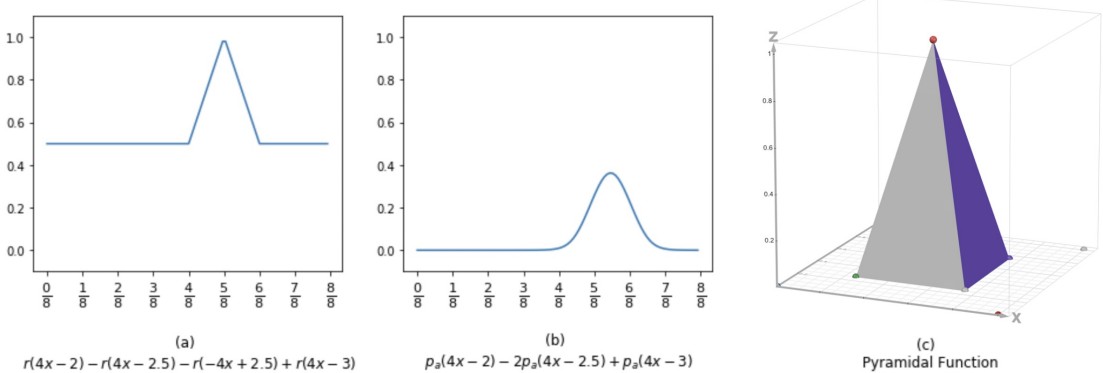

Figure 4: (a) describes the counter example $g_{n,k}+0.5$, (b) describes a perturbed Schauder built using Softplus functions $p_a(x)$ with $a = 10$, and (c) shows the pyramidal function used in Theorem 8.

