# OpenReview forum: "Schauder Bases for $C[0, 1]$ Using ReLU, Softplus and Two Sigmoidal Functions"
_TMLR — Accepted by TMLR_

### Review · Reviewer_Ewx5 · 2025-07-03

**Summary Of Contributions:**

This paper derives four versions of Schauder basis for $C[0,1]$ functions, one using ReLU, one using Softplus and two using sigmoid functions. The ReLU basis is obtained through the decomposition of the original Schauder basis into ReLU basis, and the Softplus basis is obtained through the stability property of Schauder basis and perturbation of the ReLU basis. Following the construction of these two basis, sigmoid versions of them are constructed by representing the basis as their first discrete derivatives.

**Audience:**

Yes

**Broader Impact Concerns:**

This is a foundational theoretical work, I believe broader impact does not apply here.

**Claims And Evidence:**

Yes

**Requested Changes:**

The paper itself is well-written and easy to follow. The only changes that I would request is a discussion or remark on the potential implications of this work in the context of machine learning, highlighting the relevancy of this work in the TMLR community.

**Strengths And Weaknesses:**

Strength:

1. This paper is very well written with concise and self-contained writing, which makes it easy to follow and enjoyable to read.
2. The results provide a new perspective in looking at ReLU, Softplus, and Sigmoid activation functions in neural networks.
3. The existence of Schauder basis in the form of given functions is interesting, which, potentially, could have implications in approximation theory that helps provide better explanation in the approximation power of finite dimension neural networks (lightly hinted in Section 1).

Weakness:

My main concern regarding this work is in terms of existence of audience in TMLR community. Although I personally enjoyed reading the work (and as such, I selected yes to the audience question), the subject itself is more of a mathematical piece than a topic directly relates to machine learning. I do not see any direct practical implications (algorithmic) of such derivation in machine learning community, as the results only wrap up to the existence and uniqueness of the coefficient $\alpha$. The only loose connection to machine learning that I could identify is Strength 3 I stated above. Therefore, at the current stage, it is still hard to justify the relevancy of this work in the TMLR community.

---

### Review · Reviewer_Bxph · 2025-07-27

**Summary Of Contributions:**

The authors show how to construct Schauder bases for C[0,1] using ReLU functions, and softplus functions. The proof of the existence for ReLU functions builds on a prior result that provides a Schauder basis of C[0,1] in terms of piece-wise linear functions. The proof for softplus functions follows from the result for ReLUs and a stability argument, The authors also translate these two results into the constructions of Schauder bases using the sigmoidal versions of ReLU and softplus, which are the functions that yield the ReLU and softplus upon discrete differentiation.

**Audience:**

Yes

**Broader Impact Concerns:**

None.

**Claims And Evidence:**

Yes

**Requested Changes:**

- Could the authors comment on extensions to Schauder bases of C([0,1]^n), since neural networks are most useful in high dimensions. Intuitively, it looks like the ReLU argument should go through using the n-dimensional Schauder function.
- Can the authors comment on the approximation error incurred by the truncated expansion associated to the Schauder basis, especially in comparison to results in which the ReLU functions can be chosen adaptively, as in Barron’s work [1], and follow-ups such as [2], [3], [4], [5]. If a theoretical analysis is possible, I’m expecting the bound to be much worse. This is somewhat similar to the separation results between the F1 (adaptive) and F2 (fixed features, RKHS) shown in [6].

[1] A. Barron. Universal approximation bounds for superpositions of a sigmoidal function. IEEE
Transactions on Information Theory, 39:930 – 945, 1993.

[2] L. Breiman. Hinging hyperplanes for regression, classification, and function approximation. IEEE
Transactions on Information Theory, 39(3):999–1013, 1993.

[3] J. M. Klusowski and A. R. Barron. Approximation by combinations of relu and squared relu
ridge functions with ℓ1 and ℓ0 controls. IEEE Transactions on Information Theory, 64(12):
7649–7656, 2018.

[4] G. Ongie, R. Willett, D. Soudry, and N. Srebro. A function space view of bounded norm infinite
width relu nets: The multivariate case. In International Conference on Learning Representations
(ICLR 2020), 2019.

[5] C. Domingo-Enrich, Y. Mroueh. Tighter sparse approximation bounds for ReLU neural networks. In International Conference on Learning Representations
(ICLR 2022), 2021.

[6] F. Bach. Breaking the curse of dimensionality with convex neural networks, Journal of Machine Learning Research, 2017.

**Strengths And Weaknesses:**

These are strengths of the paper:

- The paper shows clean theoretical results.

These are weaknesses of the paper:

- The proofs are not highly novel technically. The proof for ReLU functions is basically immediate, by rewriting Schauder hat functions as linear combinations of ReLUs. The more challenging part is the stability argument to obtain the result for softplus. The extensions to the sigmoidal versions are also straight-forward.
- The existence of Schauder bases using particular activations may be interesting from a theoretical standpoint, but is very removed from any kind of empirical phenomenon, because in practice, when training neural networks, the ReLU activations are learned. Hence, I would argue that the results shown are more appealing to analysts than to machine learning researchers.
- No algorithm is proposed to learn/approximate the coefficients. Is this possible? Can the authors comment?

---

> ### Author Response · Authors · 2025-09-06
> **Bounding approximation error and truncation error using Schauder basis**
>
> We would like to thank Bxph for their review comments, and helpful pointers to connect our paper to approximation results in currently active machine learning theory. We will provide a brief response here, and follow-up with an updated manuscript after consolidating all the feedback.
>
> First off, we note that for the subclass of Lipschitz functions within $C[0,1]$ we are able to show an $O(\frac{1}{n})$ approximation error using $n$ ReLU basis functions, though our current approach is based on interpolation and not series truncation. We can also show a negative result that a single layer ReLU network cannot act as a basis for multidimensional input data. We will document these results in detail as part of our updated manuscript. We will now respond to the specific comments, though we proceed in reverse order and indicate some ongoing work.
>
> We start with the last comment on approximation error, and the nice list of references documenting $O(\frac{1}{n})$ approximation error, starting with Barron’s 1993 paper [1]. We summarize these and a few other results, including our own, as follows: firstly, there are fully adaptive Kolmogorov style basis functions which support perfect representations using a fixed number of functions. Next there are semi-adaptive functions as considered in the suggested references where the form of the function is fixed, say to ReLU or some other sigmoidal function like tanh, while the inner parameters ($w, b$ in $w^Tx + b$) are allowed to be adaptive. For these semi-adaptive basis functions Barron establishes $O(\frac{1}{n})$ estimation error, and Bach [6] establishes similar bounds for classes of functions like F1. Savarese [10] looks at infinite series representations based on semi-adaptive functions and establishes $O(\frac{1}{n})$ truncation error in these cases as well. Ongie [4] links the constant in these approximations to the R-norm of the target function f. Finally, there are fixed basis functions as described in our paper, but the approximations in such cases are limited to $O(\frac{1}{n^{1/d}})$ where $d$ is the number of dimensions. Given such a fixed basis, Daubechies [8] looks at n-term approximations using the basis elements and indicates $O(\frac{1}{n})$ approximation error for univariate functions.
>
> Our $O(\frac{1}{n})$ error bound is most similar to the Daubechies result in that it uses the basis as a dictionary for an n-term approximation. Our bound is based on the first n basis functions, but the coefficients are determined by interpolation, and not by series truncation.
>
> On the question of extending the basis to d-dimensions, a natural approach is to use pyramidal functions as in Semadini [7, p.54], as they generalize the Schauder hat functions to d-dimensions. But there are some challenges in pursuing this idea. Individual basis functions have compact support under these pyramidal schemes based on the dual polytope of the unit cube (cross-polytope). Our attempts at constructing these pyramidal functions based on the ReLU function lead to dual polytopes that are “too small” and thus cause difficulties for dimensions $d > 2$ as observed by Semadini [7, p.54]. We are still investigating these possibilities for constructing a multi-layer ReLU basis for $d \ge 2$.
>
> Regarding the first question on learning the coefficients: one approach, which works for the 1-D case and is implicit in our first response, is to interpolate using grid points. We expect that standard gradient descent schemes can learn these interpolation coefficients based on sample data, particularly if we freeze the basis functions $(w_i, b_i)$.
>
> Finally, we may add that there are zero-shot learning efforts (e.g. [9] TMLR 2024) where the network is trained on multiple time series datasets. In such cases, a fixed basis may be a better choice since adaptation is not an option. In such cases, the capacity to support a basis may be part of the reason why ReLU networks are successful.
>
> References:
> [1]-[6] as in the original review.
> [7] Semadeni, Zbigniew. Schauder bases in Banach spaces of continuous functions. Springer (1982).
> [8] Daubechies, I., DeVore, R., Foucart, S. et al. Nonlinear Approximation and (Deep) Networks. Constructive Approximation 55, (2022).
> [9] Ansari et. al. Chronos: Learning the language of time series. TMLR (2024).
> [10] Savarese, Pedro, et al. "How do infinite width bounded norm networks look in function space?." Conference on Learning Theory. PMLR (2019).

---

### Review · Reviewer_nkMh · 2025-09-06

**Summary Of Contributions:**

This paper constructs several different so-called *Schauder basis* functions for the space of continuous functions on the unit interval. A basis is Schauder if, when writing an arbitrary such function $f$ as an infinite linear combination of basis elements, the coefficients are continuous linear functionals of $f$. Constructions of such bases are already known, e.g., Schauder's original construction consists of a collection of piecewise-linear hat-shaped bumps.

This paper first uses the fact that such piecewise-linear functions can be written as linear combinations of ReLUs to construct a Schauder basis of shifted and scaled ReLUs. It similarly shows that the Schauder's hat functions can be written as differences of step functions (given by differences of ReLUs) to exhibit another Schauder basis consisting of scaled and shifted step functions; these steps have the additional nice property of being "sigmoidal", that is, they tend to 0 on the left and 1 on the right.

In the second half, the paper further uses standard stability properties of Schauder bases to show that softplus functions also form a Schauder basis because they can be obtained by perturbing ReLUs. More concretely, these stability results say that if the elements of a Schauder basis are perturbed by a small enough amount pointwise, they remain a Schauder basis. Using the same idea but perturbing the step functions above instead, they construct a sigmoidal basis out of softpluses.

**Audience:**

No

**Claims And Evidence:**

Yes

**Requested Changes:**

- P. 2: "minimmum"
- Typo in Theorem 4: $\sum_i$ should be $\sum_n$
- It would be helpful to provide plots of some of the relevant activation functions

**Strengths And Weaknesses:**

**Strengths**: I'm not an expert in this area and wasn't aware of the stability result used in the second part of the paper. This result gives a clean way to extend their result for ReLU to softplus. The main novelty of the softplus results is that unlike Schauder's original construction, the constructions in Theorems 3 and 6 are smooth.

**Weaknesses**: I didn't see a convincing argument in the introduction section for why to study these questions. I can understand that maybe the ReLU result is more desirable than the Schauder hat given the practical relevance of the former, but the technical contribution here is quite limited as they just use the fact that piecewise linear functions can be written as linear combinations of ReLUs. Also while the softplus results are less trivial, these activations are less standard, and from a technical standpoint the techniques seem quite standard within approximation theory.

---

> ### Author Response · Authors · 2025-09-17
> **Summary of relevant changes**
>
> We would like to thank nkMh for their review. We have incorporated various changes in the manuscript based on their comments. We have added some new results and introductory remarks connecting our basis results to questions of direct interest to the neural network community - like approximation guarantees for finite neural networks (Theorem 7), the possible need for multilayer networks (Theorem 8), and the effectiveness of the ReLU function. We have also tried to delineate the routine and subtle portions of the basis construction, including a counter example
> $g_{n,k}(x) = r(2^nx - k) - r(2^nx - (k+\frac{1}{2})) - r(-2^nx + (k + \frac{1}{2})) + r(-2^nx + (k+1)) - \frac{1}{2}$
> which shows that arbitrary (trivial) linear combinations of piecewise linear functions, though summing to the Schauder hat function, don’t always lead to a valid basis. We also hope the reviewer finds the negative result in Theorem 8 interesting, and the non-standard combinatorial argument therein. Finally, we started work on some graph plots and we’ll update them once we consolidate any final feedback. This was a good idea, and we used it to illustrate the newly added pyramidal function as well.

---

### Author Response · Authors · 2025-09-17
**Acknowledgement of our responses**

Dear reviewers,

Thank you for your review comments. They have been very helpful, and we have incorporated various changes to the manuscript based on your comments.

We believe our result is a modest one, but fairly striking since such a basis result has not been articulated for ReLU or sigmoidal functions in the last (about) 40 years since the early universal approximation theorems of the 1980s. We believe an elegant construction hides a delicate result which doesn’t generalize easily to other (trivial) linear combinations or to multiple dimensions. On the other hand, the results point to a natural requirement for multilayer networks, and to some alternate approximation results for neural networks based on the supnorm which may help us understand neural network behavior, and the effectiveness of the ReLU function from a new point of view.

If you find our responses satisfactory, we would greatly appreciate it if you could acknowledge them with a comment. Should you have any further questions or feedback, we would be happy to address them. In particular, we welcome any additional suggestions you may have for enhancing the quality and clarity of the manuscript, especially those you consider critical. Thank you for your time.

We look forward to your responses.

Regards,
Authors.

---

### Decision · Action_Editor_Qwbf · 2025-10-20

**Recommendation:** Accept as is

**Additional Comments:**

The authors already addressed the concerns of the reviewers with an updated manuscript. The submission is acceptable in its current form.

**Audience:**

Yes

**Audience Explanation:**

While all reviewers initially expressed concern that the paper might be too mathematical and of niche interest, they all ultimately agreed that it is suitable for the TMLR audience. The central topic of understanding the fundamental approximation properties of activation functions like ReLU is a core area of interest for the learning theory community.

**Claims And Evidence:**

Yes

**Claims Explanation:**

The paper constructs several Schauder bases for the space of continuous functions on the unit interval, $C[0,1]$, using functions relevant to machine learning, namely ReLU, softplus, and their sigmoidal variants. The theoretical claims are rigorous and well-supported.

All three reviewers shared two major concerns: (1) limited technical novelty and (2) lack of relevance to ML. In particular, the construction for ReLU was seen as a straightforward extension of existing results for piecewise-linear functions. In the rebuttal, the authors provided a substantial revision that successfully addressed these concerns. In particular,
- The authors added a new "Applications" section that significantly improved the paper's motivation and relevance. This included a new theorem on approximation guarantees for finite neural networks and a negative result showing that a single-layer ReLU network cannot form a basis in multiple dimensions, thus motivating the need for multilayer networks.
- They additionally clarified technical subtleties with a new counterexample, demonstrating that their basis construction is non-trivial.
- They added a more detailed discussion about their work in the context of other results in approximation theory, comparing their fixed-basis results with adaptive-basis results like those of Barron.

These changes were considered satisfactory and we unanimously agreed to accept the paper.